# Mental Health Patient-Reported Outcomes and Experiences Assessment in Portugal

**DOI:** 10.3390/ijerph191811153

**Published:** 2022-09-06

**Authors:** Anabela Coelho, Katherine de Bienassis, Niek Klazinga, Susan Santo, Patrícia Frade, Andreia Costa, Tânia Gaspar

**Affiliations:** 1Comprehensive Health Research Centre (CHRC), Nursing Department, University of Évora, 7004-516 Évora, Portugal; 2H&TRC-Health & Technology Research Center, ESTeSL-Escola Superior de Tecnologia da Saúde, Instituto Politécnico de Lisboa, 1549-020 Lisbon, Portugal; 3Global Health and Tropical Medicine, Instituto de Higiene e Medicina Tropical, Universidade NOVA de Lisboa, 1099-085 Lisbon, Portugal; 4Organisation for Economic Co-Operation and Development, 75016 Paris, France; 5Department of Public and Occupational Health, Amsterdam UMC, Amsterdam Public Health Research Institute, University of Amsterdam, 1105 AZ Amsterdam, The Netherlands; 6Consultant Psychiatrist Centro Hospitalar de Entre Douro e Vouga, 4520-211 Santa Maria da Feira, Portugal; 7Psychiatrist, Director of Psychiatry and Mental Health Department and Integrated Responsibility Center, CHO E.P.E, 2500-176 Caldas da Rainha, Portugal; 8Nursing Research, Innovation and Development Centre of Lisbon (CIDNUR), Nursing School of Lisbon (ESEL), 1600-096 Lisbon, Portugal; 9Católica Research Centre for Psychological, Family and Social Wellbeing, Faculdade de Ciências Humanas, Universidade Católica Portuguesa, 1649-023 Lisbon, Portugal; 10Instituto de Saúde Ambiental (ISAMB), Faculdade de Medicina, Universidade de Lisboa, 1649-028 Lisbon, Portugal; 11Digital Human-Environment Interaction Labs (HEI-LAB), Universidade Lusófona de Humanidades e Tecnologias, 1749-024 Lisbon, Portugal

**Keywords:** patient-reported outcome measures (PROM), patient reported experience measures (PREM), health system, patient outcomes, mental health, community health

## Abstract

Mental ill-health is increasingly recognized by policymakers for its significant human and economic toll. The main objective of this study is to capture patient-reported outcomes and experiences on mental health care in Portugal using methods developed for international benchmarking purposes, such as the OECD Patient-reported Indicators Surveys. The study included 397 participants, 247 (62.2%) women, divided into four age groups: ages 16–24 years, ages 25–44 years, ages 45–65 years, and ages 66 years or older. The data collection procedure and analysis followed the OECD PaRIS Mental Health Working Group 2021 protocol allowing subsequent comparability with data from other OECD member countries. Findings on the WHO-5 Well-Being Index showed that women manifest a lower score in well-being following mental health care services use. This finding may be, at least in part, explained by the study population (mental health services users), including individuals with clinical depression which is more frequently observed in women. In terms of the level of satisfaction with treatment (provided by nurses, doctors, phycologists, etc.) the response “Yes, definitely” varied from 67% of answers regarding “time spent by care providers”, 76.3% “involvement in decisions” to 79.7% regarding “clarity of explanations” and 84.4% regarding the item courtesy and respect. This study shows the feasibility of implementing and using patient-reported metrics (PROM and PREM) in mental health services in Portugal. The study results generate useful clinical information to help meet the expectations and needs of patients, contributing to a continuous improvement of mental health community services.

## 1. Introduction

Health systems, around the world, are working to find ways to innovate mental health service delivery and policies to improve quality of care and patient safety, while dually ensuring efficiency and sustainability. According to recent OECD data, many countries consider that their mental health care systems are inadequate [1]. This is a problematic trend, as mental health is a vital component of individual well-being as well as social and economic participation.

The costs of mental ill-health from the economic and social perspectives are significant. The OECD estimates that up to 13% of total health spending is directed to mental health services [2]. Other social costs, related to employment rates and productivity of people living with mental health problems, reach almost 1.6% of GDP in EU countries [2]. This significant economic impact is due to costs of under- and unemployment, low productivity related to mental ill-health, and expenditure on social support for professional disability.

According to Statista Research, Portugal invested, approximately, 136.2 million Euros in mental health hospitals in 2019 [3], however, 80% of the total expenses are due to hospitalizations and emergency consultations in mental health [4].

Issues related to health and mental illness are complex in that they affect people’s individual lives, their relationship with others, and their surroundings [5,6].

Understanding the impact of mental health care on the Portuguese population is still limited and data are scarce. In Portugal, despite the high prevalence of psychiatric disorders, existing data suggest that a significant proportion of people do not receive adequate mental health care. Data from the epidemiological study on mental health [7] in Portugal show that the treatment gap, between those who need mental health care and mental health care recipients was 64.9% for moderate disorders and 33.6% for more severe disorders. Likewise, the results showed that less than half of people with a psychiatric disorder started treatment in the first year after the onset of symptoms. Findings from the Health Regulatory Entity show that there are barriers in accessing services which may be attributable to factors such as geographic location, for example [8].

Portuguese epidemiological studies on mental health show that the prevalence of psychiatric disorders was present in more than a fifth of the people interviewed (22.9% of the sample) in the 12 months before the study [7]. This prevalence is the second highest at the European level, with a value almost equal to that of Northern Ireland, which occupies the first place. In this respect, Portugal differs significantly from all other southern European countries, which, without exception, have a much lower prevalence than northern countries [9,10].

Additional findings from the Portuguese epidemiological study on mental health show that the degree of severity of psychiatric disorders as a whole are mostly distributed among groups of mild and moderate severity (31.9% and 50.6%, respectively), with severe cases corresponding to only 17.5% of all cases. Almost 5% of the general population has a severe psychiatric disorder, 11.6% a moderately severe disorder, and 7.3% a mildly severe disorder. Regarding the adequacy of care, the study showed that 31.4% received adequate care from general and family medicine physicians and 48.6% received specialized care in mental health. Data show that diagnosis and treatment are made 2 and 3 years later, on average, in cases of panic disorder and generalized anxiety disorder; in cases of depressive disorders: 3, 4, and 6 years, respectively, in dysthymia, major depression, and bipolar disorder [7].

In a study carried out by Gouveia M. et al., the cost and burden of schizophrenia in mainland Portugal in 2015 were estimated, concluding that the social impact of schizophrenia in Portugal is mainly due to the morbidity generated, costing a total of €436.3 million annually, about 0.24% of the gross domestic product. Direct costs represent 0.6% of all health expenditures in 2015, while total costs (direct and indirect) represent 2.7% of health expenditure [11].

Patient-reported measures can enhance the quality of care provided to individuals diagnosed with a mental condition [12] and are a critical tool for improving policy and practice in mental health care.

The OECD encourages countries to systematically adopt the monitoring of indicators related to Patient-Reported Outcomes (PROM) and Patient-Reported Experiences (PREMs). This data should be collected in such a manner that local institutions, regions, or countries can use information collected for strategic and analytical purposes, supporting macro decisions at the level of health policies, but also meso- and micro-decisions at the level of quality and safety of healthcare provision and good clinical practices [13]. Harmonized data collection and reporting practices at the national level can be used for the purposes of international benchmarking. This shared international PROM and PREM has the potential to collect user outcomes and experiences in a more ecological way and can help position service users at the heart of the mental health system [14].

With the Portuguese Decree-Law n° 113/2021 of 14 December which establishes the rules for the organization and functioning of mental health services, and the Regulation (EU) 2021/241 of the European Parliament and of the Council of the European Union of 12 February 2021 that establish the Recovery and Resilience Facility, it is expected that the mental health reform in the National Health Service should be concluded by 2026 with all the guiding principles of the organization, management, and evaluation of mental health services established.

It is expected that Mental Health Services come to be transformed into Integrated Responsibility Centers (CRIs) to improve the efficiency and the quality of care provided, with incentives given to professionals.

An alignment is also expected with the main national and international strategic instruments in terms of the legal rights of persons with disabilities, established by the Convention on the Rights of Persons with Disabilities and adopted by the ONU, by the principles 17 and 18 of the European Pillar of Social Rights, the Strategy on the Rights of Persons with Disabilities 2021–2030, of the European Commission, and by the National Strategy for the Inclusion of Persons with Disabilities 2021–2025.

There are more than enough grounds to support the widespread implementation of PROMs/PREMs related to ethical issues (since it is a fundamental right of health service users to be able to express their lived experience as a form of active participation), to clinical issues (that can be improved with the integration of the patient’s view of the care received once it is demonstrated that there are discrepancies in the assessments of health status and needs carrying out by doctors and patients) and institutional issues (once we measure what really matters, we introduce institutions to users and they become able to adjust care in a more personalized and adjusted way) [14].

The principal aim of this study is to capture patient-reported outcomes and experiences in mental health care in Portugal.

## 2. Materials and Methods

### 2.1. Study Design

The current study was a prospective cross-sectional study, involving participants that were at two different stages. Data were collected from participants at the beginning of their treatment and from participants during the treatment process.

### 2.2. Participants

The study includes 397 participants, 126 participants were at an early stage of treatment, and 271 were at the end or at the continued stage of treatment.

Of the total participants in the study, 247 (62.2%) were women, divided into four age groups: 5.6% aged 16–24 years, 31.2% aged 25–44 years, 51.1% aged 45–65 years, and 12.1% aged 66 years or older.

### 2.3. Instrument

The instruments used follow the protocol of the OECD PaRIS Mental Health Working Group: 2021 (Paris, France) [15].

The WHO-5 Well-Being Index consists of 5 items with 6 responses categories ranging from “All the time” to “At no time” with an internal consistency (α = 0.87), range 1 (less well-being) to 25 (more well-being). The Major Depression Inventory (ICD-10) is recommended if the raw score is lower than ‘13’, or if the patient answered ‘0’ or ‘1’ to any of the 5 items. A score of less than 13 reveals poor well-being and is an indication to test for depression according to ICD-10.

In relation to treatment satisfaction, a rating scale consisting of 4 items with 4 responses categories ranging from “Yes, definitely” to “No, definitely not” with an internal consistency (α = 0.84) was used. Regarding the WHO-5 scale and the scale of satisfaction with treatment, in the first phase the analysis was performed item by item and later the items of each scale were added.

In addition to the WHO-5, two items were included on life satisfaction and finding meaning in life, and four PREMS items on whether the treatment contributes to patient’s well-being and life satisfaction. These questions were measured on a scale from 0 to 10 being 0 (not at all) and 10 (completely).

The participants were asked to report back to the last week.

### 2.4. Procedure

The data collection procedure followed the OECD PaRIS Mental Health Working Group 2021 protocol allowing subsequent comparability with data from other European and OECD countries [15].

The protocol followed several phases. In the first phase, experts in the area of mental health, such as doctors, nurses, and psychologists, were gathered, and the translation of the instruments into the Portuguese language and options in the data collection procedure were discussed and reflected upon. Following meetings with experts, a data collection protocol integrating the information from the original protocol and specificities of the national context was developed by the research team.

Hospitals were invited to participate in the study, awareness-raising meetings were held, with the hospital’s involvement, and doubts were clarified with the research team. Eight participating hospitals submitted the project to the respective ethics committees, and, after the necessary approvals, the data collection process began. 

The questionnaire was distributed via an online survey sent by the research team to each participating hospital. Data were collected regarding two different stages of care: 31.7% (*n* = 126) of the participants were at an early stage of treatment and 68.3% (*n* = 271) at a continued stage of treatment. The study is cross-sectional, the participants who responded in the initial phase of their treatment and those who responded in the final phase/during treatment are not the same. In this way, we present the analysis of the participants altogether and we also analyze the comparison between the groups (participants who responded at the beginning of their treatment and those who responded at the end or during the treatment process).

Data were collected in three different contexts: 69.2% in hospital outpatient setting/outpatient consultation or day hospital; 6.3% in the hospital inpatient setting and 33.5% in community outpatient setting.

At the end of the data collection process, the research team held an event to present the overall results and drafted a specific report for each of the participating hospitals with conclusions and recommendations.

### 2.5. Ethical Consideration

This cross-sectional study had been approved by the Ethics Committee of all the health care units (N.° 112/CES/JAS; Ref.^a^ 028/CLPSI/2021). 

## 3. Results

### 3.1. Patient-Reported Outcomes Measures (PROM)

Regarding the question “how satisfied are you with life as a whole these days?” in a range from 0 (not at all satisfied) to 10 (completely satisfied) we obtained a mean of 6.02 and a standard deviation of 2.87.

Regarding the question “to what extent do you feel the things you do in your life are worthwhile?” in a range from 0 (not at all satisfied) to 10 (completely satisfied) we obtained a mean of 6.98 and a standard deviation of 2.79.

Regarding WHO-5 Well-Being Index in a range 0 (less well-being) to 25 (more well-being), we obtained a mean of 11.67 and a standard deviation of 5.81.

Regarding Treatment satisfaction in a range 4 (higher satisfaction) to 16 (lower satisfaction), we obtained a mean of 5.10 and a standard deviation of 1.85.

Table 1 presents the frequencies of the response about the well-being of the participants; the less positive aspects of well-being are related to daily life is filled with things with interest and feeling calm and relaxed.

### 3.2. Patient-Reported Experiences Measures (PREM)

In relation to Treatment satisfaction (Table 2), a high level of satisfaction was found in all assessed areas. A lower level of satisfaction was found in relation to care providers spending enough time with patients and care providers explaining things in a way that was easy to understand. 

Regarding the question “Does the treatment contribute to your well-being and satisfaction with life?” on a range of 0 (not at all) to 10 (completely) we obtained a mean of 7.83 and a standard deviation of 2.42.

### 3.3. Comparison of Groups

Comparing groups on the domains of well-being and satisfaction with the treatment we found that there are statistically significant differences between men and women, with women showing a lower well-being index and satisfaction with treatment when compared to men. There are no statistically significant differences between the age groups. Regarding the moment of assessment, statistically, significant differences are found in terms of satisfaction with treatment, with a higher level of satisfaction at the moment of assessment (early vs. late stage) during treatment (Table 3).

The Linear regression aims to analyze the explanatory value of the predictors variables “life satisfaction”, “life is worth” and “treatment satisfaction” in the participants’ global well-being (here measured by the WHO-5 well-being Index). Control was performed including gender and age variables.

The linear regression presented in Table 4 reveals that in the male gender, the perception that life has worth and treatment satisfaction contributes positively to well-being and explains 42% of well-being (R^2^ = 0.42). Age and life satisfaction do not predict statistically significantly global well-being.

## 4. Discussion

The PROMs and PREMs ensure that patients, clinicians/institutions, and governments can use information collected for individual (micro), analytical (meso), and strategic (macro) purposes.

Considering the macro decisions, at the level of health policies it is known that health systems strive for greater sustainability and equity regarding mental health care services, however, 67% of those who need mental health care had difficulties accessing care [16]. The inclusion of PROMs and PREMs in the evaluation of health care interventions helps decision-makers to adjust their policies to the population’s needs, improving, on one hand, the contracting of health care services and, on the other hand, the measurement of service’s performance that is desired to be aligned with the quality standards [17].

In Portugal, PROMs and PREMs can be a useful tool for the evaluation of new mental health policies or strategies, such as the Integrated Responsibility Centers (CRIs), and used for international benchmarking purposes, such as OECD Patient-reported Indicators Surveys (PaRIS Mental Health Pilot Data Collection), however, it is known that public benchmarking can induce deviant practices (patient selection) in order to obtain, not the real, but the best picture published [17].

PROMs and PREMs also have value in the meso (clinicians/institutions) and micro (patients) levels of the health system once it will drive care to quality and to a more patient-centered service.

Despite the general high performance of the findings of this study, there is still room for improvement in relation to human resourcing and workflow planning to provide mental health specialized care and the organization of these units.

Improved time spent with patients is important for improving the shared decision-making process, i.e., exchanging information, preferences, and values about treatments, explicit reasoning about choices, and achieving agreement about the treatment plan between patients and providers [17,18].

In a recent study of a representative sample of the Portuguese population [19], it was demonstrated that shared decision-making is more acceptable to better-educated patients in the problem-solving component and to people who are younger, higher educated, and employed, in the decision-making component [18]. Particular attention should be paid to those who have lower health literacy because people with low literacy skills report four times more fair or poor health compared to those who have high literacy skills [20]. The authors refer to nurses and physicians’ training in shared-decision making with their patients as potentially useful as well as the inclusion of shared-decision making in practice guidelines for preference-sensitive healthcare decisions.

Health Professionals, such as nurses, medical doctors, and psychologists, must be able to free themselves from their own goals, respect each person in his or her uniqueness, look far beyond his or her disease and help them to be both knowledgeable and critical of all his/her therapeutic process [21]. Similarly, it is essential to empower the person with experience in mental health in a process of recognition, creation, and use of resources and instruments that translate into an increase in the efficiency of their participation [22]. Participation is an essential condition for identifying needs, planning measures, and evaluating services.

Findings from Portugal are comparable to international findings measuring PROMs and PREMs in mental health care, including the OECD PaRIS pilot data collection on mental health which included 15 data sources from 12 countries, collected over the course of 2021. Findings from this work show that an average of 85% of hospitalized patients and 88% of individuals receiving community mental health services reported being treated with courtesy and respect by their care providers among sites/countries that were able to submit data. An average of 78% of hospitalized patients and 88% of individuals receiving community-based mental health services felt that their care providers explained things in a way that was easy to understand. The share of mental health service users who felt satisfied with their involvement in their treatment decisions was 81% for hospitalized patients and 87% for individuals receiving community health services [14], however it is enhanced by several studies that show that patient satisfaction usually seems to increase when PROMs are used for clinical purposes [17].

The WHO-5 Well-Being Index is a short self-reported measure of current mental well-being. The finding that women manifest a lower score in well-being may, in part, be explained by the population being studied (mental health services users), which includes individuals with clinical depression, a syndrome more frequent in women [23]. The evidence is clear regarding complex problems (e.g., depression) suggesting that PROMs, if ambiguous, should be used with other decision-making tools, such as disease management plans and clinical pathways [17].

Portuguese population inequalities in well-being outcomes have been previously documented [24]. In the last OECD report about well-being programs, it was found that every day, women work 25 min longer than men when both paid and unpaid work (such as housework and caring responsibilities) are considered [25]. Although there has been progress in gender equality in Portugal in the last years [24,26], women are much more likely than men to do cooking and housework every day for at least 1 h. This gender gap is among the widest in the European Union [24].

In recent years, we have seen an important paradigm shift in mental health in Portugal. There is a clear improvement in existing legislation while seeing consistent and innovative alternatives emerging. However, there are still challenges in various domains and the mainstreaming of mental health services.

Initial findings in Portugal have found that improved participation of the person with experience in mental illness has a positive effect on their psychosocial rehabilitation process. Therapeutic teams, involving nurses, physicians, psychologists, etc. should be able to develop quality interventions supported by a biopsychosocial model allowing people with experience in mental illness to achieve greater autonomy and greater participation [27]. The health professional must be able to look genuinely at the other, demonstrating interest and understanding, encouraging his processes of change, while giving him the freedom to be himself, free and autonomous in his choices and decisions [28]. The therapeutic team should discuss the results of PROM with patients because this feedback recognizes them as a partner in the clinical process, and it is proven by several studies carried out in mental health services that this will improve reflection on the practices and therapeutic outcomes [29].

PROMs and PREMs can also be useful instruments as they raise awareness of how involved the patients feel in each decision; therefore a ‘bottom-up’ (micro and meso) approach combined with a “top-down” (macro) guidance appears to be the best strategy for improving health service performance and promoting a more efficient clinical practice [17].

## 5. Limitations and Suggestions for Future Research and Actions

This study has a few limitations that should be highlighted for future research. Regarding the level of satisfaction with treatment, it might be useful to consider the different methodologies used to obtain the responses to the questionnaires to minimize the possibility of a component of social desirability bias in the answers. For future studies and actions, we recommend exploring all the answers “yes, to some extent”. Those who responded might help, in the future, to point out which aspects they felt as helpful and those which were not satisfying to experience as a mental health service user. The study integrates participants from a clinical sample. In future studies, the perception of a non-clinical population could also be included.

## 6. Conclusions

Ecological analysis (macro, meso and micro level) of the health systems, healthcare organizations/professionals, and patients are valuing PROMs and PREMs in different areas, such as personalized care, shared decision making, quality improvement, efficiency, and transparency.

PROM and PREM can be a useful tool for the evaluation of new mental health policies or strategies, such as the Portuguese Integrated Responsibility Centers (CRIs) in mental health, and used for international benchmarking purposes, such as OECD Patient-reported Indicators Surveys (PaRIS Mental Health Pilot Data Collection), however, a political commitment and a stable budget are required to set up an official structure to ensure validated strategies and results from PROMs and PREMs.

The evaluation of the quality of services has a very important component: patient satisfaction. PROM and PREM provide this response and allow health care services to capture patient-reported outcomes and experiences in mental health care. These measures meet the expectations and needs of patients, contributing to the continuous improvement of those services, however, these measures, in Portugal, should be embedded in a global quality assurance policy in order to improve, for example, the lower level of satisfaction of our patients regarding the time spent with them and regarding their involvement in treatment decisions.

Patients are considered “experts” of their disease and partners in their own medical decision-making process, therefore PROMs and PREMs can be used to support shared clinical decisions and to promote patient-centeredness of care, by improving the communication between the patient and the health care provider. The study suggests that patient well-being should be understood in a multidimensional approach, associated with satisfaction with the treatment and in relation to the patient’s life worth. This aspect reinforces that the promotion of patient well-being requires an integrated approach with intervention at the level of hospital treatment and psychosocial reintegration into the community.

## Figures and Tables

**Table 1 ijerph-19-11153-t001:** Frequencies of the response about the well-being of the participants.

WHO-5 Questions	All the Time%	Most of the Time%	More than Half of the Time%	Less than Half of the Time%	Some of the Time%	At No Time%
I have felt cheerful and in good spirits	5.8	29.8	19.1	25.4	14.6	5.3
I have felt calm and relaxed	8.8	22.4	14.6	27.7	10.7	5.8
I have felt active and vigorous	11.3	24.4	16.4	26.5	12.8	8.6
I woke up feeling fresh and rested	14.9	27.6	15.4	21.9	14.9	5.3
My daily life has been filled with things that interest me	7.3	26.0	14.1	23.9	19.4	9.3

**Table 2 ijerph-19-11153-t002:** Frequency in relation to treatment satisfaction.

During the Course of Your Treatment:	Yes, Definitely %	Yes, to Some Extent %	No, Not Really %	No, Definitely Not %
Did your care providers treat you with courtesy and respect?	84.4	13.1	1.5	1.0
Did your care providers spend enough time with you?	67.0	27.6	3.6	1.8
Did your care providers explain things in a way that was easy to understand?	79.7	17.0	1.8	1.5
Did your care providers involve you as much as you wanted to be in decisions about your care and treatment?	76.3	20.4	2.8	0.5

**Table 3 ijerph-19-11153-t003:** Descriptive Statistics and Comparison analysis according to sex, age, and assessment moment.

	Descriptive Statistics				*t*-Test & Significance
	$\bar{x}$	SD	$\bar{x}$	SD	
	Female		Male		
WHO-5	10.32	5.67	13.89	5.37	*t* = −6.29, *p* = 0.001
Treatment satisfaction	4.95	1.76	5.35	1.97	*t* = −2.00, *p* = 0.04
	16–44		45 or more		
WHO-5	12.16	5.47	11.37	6.00	*t* = 1.34, *p* = 0.19 (n.s.)
Treatment satisfaction	5.12	1.77	5.08	1.90	*t* = 0.22, *p* = 0. 83 (n.s.)
	Initial assessment		Assessment during treatment		
WHO-5	11.26	6.38	11.85	5.53	*t* = 0.94, *p* = 0.35 (n.s.)
Treatment satisfaction	4.81	1.35	5.22	2.03	*t* = 2.05, *p* = 0.02

**Table 4 ijerph-19-11153-t004:** Linear regression model to study well-being (WHO-5).

	B	Error	Β	T	*p*
(Constant)	−0.057	0.255		−0.224	0.823
Gender (1 = female)	0.476	0.094	0.199	50.066	0.000
Age (2 groups)	−0.132	0.093	−0.055	−10.419	0.157
Life satisfaction	0.013	0.017	0.033	0.773	0.440
Life worth	0.209	0.020	0.502	100.216	0.000
Treatment satisfation	0.052	0.023	0.109	20.324	0.021

R^2^ = 0.42; F = 57.207 (5/391), *p* < 0.001. Dependent variable = well-being (WHO-5).

## Data Availability

The data presented in this study are available on request from the corresponding author. The data are not publicly available due to ownership belonging to the institutions where the study was conducted.

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
