# Peer review of "Mental Health Patient-Reported Outcomes and Experiences Assessment in Portugal"

_ijerph, 2022, doi:10.3390/ijerph191811153_

Round 1

Reviewer 1 Report

The manuscript is clear and relevant for the field of Mental health, is presented in a well-structured manner.

The references that sustain the paper are recent or relevant publications.

The methods and tools are described with details, allowing other researchers to reproduce the results. The conclusions are consistent with the evidence and arguments presented.

I recommend minor revision

The English language can be improved. For exemple:

Check the translation in the abstract (See line 25) referring to ”The principal objective” , we think the correct way will be “the main objective”; (See line 69) “between those how need mental health care and those that get it” we think that it is “between those who need mental health care and those that get it”

Review references 2; 6 e 25. (Complete with “Available online in” and “Assessed on …”

Author Response

Dear Reviewer, first of all, we would like to thank you for the pertinent and valuable comments and efforts toward improving our article. All the suggestions were considered and integrated into the final version.

Reviewer 2 Report

Dear Authors and Editors,

please find review in the attachment.

Author Response

Dear Reviewer, we would like to thank the reviewers for their thoughtful comments and efforts toward improving our article. In the following, we highlight the general concerns of the reviewer and our effort to address these concerns.

  1. What is the main question addressed by the research? The aim of this study is to present patient-reported outcomes and experiences of mental health care in Portugal. Suggestion for title: Change “Portuguese Case Study" to “Portuguese survey" or study, or “Mental Health Patient- Reported Outcomes and Experiences Assessment in Portugal” because formulation "case study" was typically used in describing different research design. This is research conducted on large sample, and do not present case study.

Response 1: Thank you for this suggestion. We update the title as recommended

  1. Do you consider the topic original or relevant in the field, and if so, why? & 3. What does it add to the subject area compared with other published material?

The topic is relevant in the field of mental health services. Paper presenting Portuguese research on large clinical sample, what can be of interest to other researchers conducting similar population, or national research.

Response 2: The authors would like to thank the reviewer for this positive statement.

  1. No Point 3
  2. What specific improvements could the authors consider regarding the methodology?

Ln.130. The current study was a prospective transversal study... In this section provide information on time-points, and in Participants section, how many participants were included in each measurement time-point. Done

For WHO-5 specify to which time period the questions refer (last week, last month, or what period of time?). Also state how the total result or wb index is calculated. For the treatment satisfaction also. Because, later, in Results section (Ln 207...) overall result was presented with a single score for well-being index and for satisfaction with treatment. Done

Ln 140. “...6 response hypotheses" change to “...6 response categories..." Done

Ln 142 same change as above Done

Ln 141 and 143  omit the word optimal Done

Ln 165 “was collected" change to “questionnaire was distributed..." Done

Results:

If this is prospective research, and if these are results from two separate measurements on two samples (two time- point, different participants) results from two time points need to be compared. And only if there is no difference, data can be presented altogether from two measurement time-points. Otherwise, if there is a difference between the measurements, than results from two measurements need to be presented separately. Answered Below

Ln 196 & 197: Regarding the question "Does the treatment contribute to your well-being and satisfaction with life? "on a range of 0 (not at all satisfied) to 10 (completely satisfied) we obtained...” The end points of answering scale is questionable. If question is: Does the treatment contribute to your well-being and satisfaction with life? Adequate answer is Not at all to Completely. Without word “satisfied" because question is about contribution TO satisfaction, not about degree of satisfaction. The described scale is adequate answering scale for the question How are you satisfied with... Done

Suggested changes in Discussion:

Ln. 219-222: it is recommended to describe what the results mean, rather than repeating the numbers from the Results section. Discussion section gives discussion about results, and comparison to findings from literature, so discussion section needs to be improved. Done

 Ln. 238 correct the word phycologist Done

Ln 282. in 5. Limitations section: it will be beneficial to point out limitation in drawing conclusion due to clinical sample. Current mental health service users only. We cannot conclude on mental health service in general because we do not know satisfaction of those who drop out or stop using service or are unable to use. Done

Response 4: The authors would like to thank all the comments and suggestions that we genuinely accept and include in the text in order to clarify the identified doubts. However we would like to reinforce that this study is a cross-sectional study, the participants who responded in the initial phase of their treatment and those who responded in the final phase/during treatment are not the same. It is not a longitudinal study, in which the same participants would respond at both times. In this way, we present the analysis of the participants altogether and we also analyze the comparison between the groups (participants who responded at the beginning of their treatment and those who responded at the end or during the treatment process). All this clarification was included in the paper.

  1. Are the conclusions consistent with the evidence and arguments presented and do they address the main question posed?

In 6. Conclusions:

Ln 291: rewrite the beginning of the sentence, indicating that you are talking about levels (macro, mezzo and micro) of health care or levels of what (society, health system, health care,...)?

Conclusion section lacks the conclusion on the obtained results and answer to aim of the research which is patient-reported outcomes, and experiences of mental health care in Portugal.

 Response 5: Once again the authors would like to thank the comment and express their commitment to the revision done in this section in order to be more clear and more precise.

  1. Are the references appropriate?

Yes, references are appropriate, and can be improved.

It is advisable to improve Introduction section with the following article:

-             Roe D, Slade M, Jones N. The utility of patient-reported outcome measures in mental health. World Psychiatry.

2022 Feb;21(1):56-57. doi: 10.1002/wps.20924. PMID: 35015343; PMCID: PMC8751576.ve the Introduction

Highly recommend to consider these articles in discussing own research and results:

-             Krägeloh CU, Czuba KJ, Billington DR, Kersten P, Siegert RJ. Using feedback from patient-reported outcome measures in mental health services: a scoping study and typology. Psychiatr Serv. 2015 Mar 1;66(3):224-41. doi: 10.1176/appi.ps.201400141. Epub 2014 Dec 1. PMID: 25727110.

-             Desomer, A., Van den Heede, K., Triemstra Mattanja, T., Paget, J., De Boer, D., Kohn, L. and Cleemput, I., 2018. Use of patient-reported outcome and experience measures in patient care and policy.

 Response 6: The authors are sensible with the involvement of the reviewer in order to improve this paper and touched with all the help provided. All the suggested articles were read and included in the introduction and discussion section.

  1. Please include any additional comments on the tables and figures.

All tables: Decimal numbers in table are not correctly written (check the English grammar and style and correct writing the decimal numbers throughout the manuscript). Done

Ln.188. Missing Table title! Done

Table 3. Overall score from WHO-5 questionnaire was not correctly calculated. See the WHO-5 questionnaire instructions for calculating and presenting overall score from 5 items. Done

In Table 3. Present exact p value for all tests (for non-significant as well) Done

Ln 209: more precisely, statistically describe what was criterion variable, what variables were included as predictors. Done

The title of Table 4. — linear regression model to study well-being. Change in a way to be clear what is criterion variable in this regression. This model was used to explain the variance of the criterion variable, what need to be clear from the Table title. Done

Here is introduced variable name: "Treatment well-being", while before was "Treatment satisfaction" or "satisfaction with treatment" — need to be aligned with the Methodology section and consistently use the same names of variables or constructs throughout the manuscript. Done    

Response 7: The authors would like to thank the reviewer again for the important comments with which we absolutely agree. We have done all the updates and clarifications proposed.

Round 2

Reviewer 2 Report

Suggested changes were included in manuscript, and overall content is improved, thus recommendation is to accept manuscript for the publication.